# Acid-treated *Staphylococcus aureus* induces acute silkworm hemolymph melanization

**Yasuhiko Matsumoto** [iD] *, **Eri Sato, Takashi Sugita**

Department of Microbiology, Meiji Pharmaceutical University, Tokyo, Japan

* ymatsumoto@my-pharm.ac.jp

## Abstract

The skin microbiome maintains healthy human skin, and disruption of the microbiome balance leads to inflammatory skin diseases such as folliculitis and atopic dermatitis. *Staphylococcus aureus* and *Cutibacterium acnes* are pathogenic bacteria that simultaneously inhabit the skin and cause inflammatory diseases of the skin through the activation of innate immune responses. Silkworms are useful invertebrate animal models for evaluating innate immune responses. In silkworms, phenoloxidase generates melanin as an indicator of innate immune activation upon the recognition of bacterial or fungal components. We hypothesized that *S. aureus* and *C. acnes* interact to increase the innate immunity-activating properties of *S. aureus*. In the present study, we showed that acidification is involved in the activation of silkworm hemolymph melanization by *S. aureus*. Autoclaved-killed *S. aureus* (*S. aureus* [AC]) alone does not greatly activate silkworm hemolymph melanization. On the other hand, applying *S. aureus* [AC] treated with *C. acnes* culture supernatant increased the silkworm hemolymph melanization. Adding *C. acnes* culture supernatant to the medium decreased the pH. *S. aureus* [AC] treated with propionic acid, acetic acid, or lactic acid induced higher silkworm hemolymph melanization activity than untreated *S. aureus* [AC]. *S. aureus* [AC] treated with hydrochloric acid also induced silkworm hemolymph melanization. The silkworm hemolymph melanization activity of *S. aureus* [AC] treated with hydrochloric acid was inhibited by protease treatment of *S. aureus* [AC]. These results suggest that acid treatment of *S. aureus* induces innate immune activation in silkworms and that *S. aureus* proteins are involved in the induction of innate immunity in silkworms.

## Introduction

The human skin microbiome maintains the skin environment to prevent the onset of inflammatory skin diseases such as folliculitis and atopic dermatitis [1–3]. *Staphylococcus aureus*, a gram-positive bacterium on human skin and in the nasal cavity, causes severe systemic infections such as sepsis and inflammatory skin infections such as folliculitis and atopic dermatitis [4–6]. *S. aureus* enters the hair follicles and proliferates, which induces immune responses and inflammation resulting in folliculitis [2, 5]. *S. aureus* peptidoglycan, a cell wall component, and proteins such as lipoproteins induce innate immune activation and inflammation [6, 7].

**Data Availability Statement:** All relevant data are within the paper and its Supporting Information files.

**Funding:** This project was partly supported by the Kose Cosmetology Research Foundation [No. 711]

to YM, the Japan Society for the Promotion of Science (JSPS) [JP23K06141 for Scientific Research ©] to YM, Japan Agency for Medical Research and Development/Japan International Cooperation Agency (AMED) [JP22wm0325054, JP22fk0108553] to YM, and Research and implementation promotion program through open innovation grants [JPJ011937] to the consortium where YM serves as a member from the Project of the Bio-oriented Technology Research Advancement Institution (BRAIN). The funders had no role in the study design, data collection, data analysis, decision to publish, or preparation of the manuscript.

**Competing interests:** The authors have declared that no competing interests exist.

Understanding the mechanism underlying the induction of host immunity by *S. aureus* may contribute to the prevention and treatment of inflammatory skin diseases.

*Cutibacterium acnes*, a gram-positive bacterium on the human skin, is a causative agent of inflammatory skin diseases, such as acne vulgaris [1, 8, 9]. Acne vulgaris is an inflammatory skin disease that affects hair follicles and sebaceous glands [9]. *C. acnes* interacts with *S. aureus* in hair follicles [5]. Various factors such as lipase and propionic acid, which slightly acidify the skin surface, are produced by *C. acnes*, [10, 11]. Lipase released by *C. acnes* produces free fatty acids from sebum [12]. *C. acnes* also secretes short-chain fatty acids such as propionic acid to lower the pH of the skin environment [13]. The effects of these acidic substances associated with *C. acnes* on the induction of innate immunity by *S. aureus*, however, remain unknown.

Innate immune activation occurs in multiple immune cells through several pattern-recognition proteins [14, 15]. Therefore, the use of individual animals is desirable for evaluating the activation of immunity. Experiments using a large number of mammals such as mice trigger ethical issues from the perspective of animal welfare [16, 17]. The silkworm, an invertebrate, has benefits as a model animal to overcome these ethical issues [16, 18, 19]. Moreover, experimental systems using silkworms have been established to evaluate the innate immune activation by pathogenic microorganisms [20–22]. The melanization response is an innate immune mechanism of insects, including silkworms [23–25]. When pathogens enter the silkworm body, the silkworms produce melanin in the hemolymph to coagulate the pathogens and repair the wound [20, 25, 26]. Melanization and immune responses via the Toll pathway are mediated by the same signaling cascades [23–25, 27]. Therefore, silkworm hemolymph melanization is a useful indicator for evaluating the induction of innate immunity by pathogens such as *Cutibacterium acnes*, *Porphyromonas gingivalis*, and *Candida albicans* [20–22].

In this study, using silkworm hemolymph melanization as an indicator of innate immune activation, we found that treatment with *C. acnes* culture supernatant enhanced the immune-inducing activity of *S. aureus*. Moreover, treatment with acidic substances such as propionic acid, acetic acid, lactic acid, and hydrochloric acid enhanced the immune-inducing activity of *S. aureus*. These findings suggest that acidification is involved in immune induction by *S. aureus*.

## Materials & methods

### Reagents

Gifu anaerobic medium agar was purchased from Nissui Pharmaceutical Co., Ltd. (Tokyo, Japan). Tryptic soy broth was purchased from Becton Dickinson (Franklin Lakes, NJ, USA). Protease K was purchased from QIAGEN (Hilden, Germany). Propionic acid, acetic acid, lactic acid, and hydrochloric acid were purchased from FUJIFILM Wako Pure Chemical Corporation (Osaka, Japan).

### Culture of bacteria

*C. acnes* ATCC6919 and *S. aureus* Newman strains were used in this study. The *C. acnes* ATCC6919 strain was spread on Gifu anaerobic medium agar and incubated under anaerobic conditions at 37˚C for 3 days [22]. The *S. aureus* Newman strain was spread on tryptic soy broth agar and incubated under aerobic conditions at 37˚C for 1 day [22]. To prepare the *C. acnes* culture supernatant (CS), *C. acnes* ATCC6919 strain ($8 \times 10^9$ cells) was added to 10 mL of tryptic soy broth + 2% glucose medium and incubated at 37˚C for 6 days under anaerobic conditions.

### Silkworm rearing

The silkworm rearing procedures were previously described [28]. Silkworm eggs (Hu Yo × Tukuba Ne) were purchased from Ehime-Sanshu Co. Ltd. (Ehime, Japan), disinfected, and hatched at 25˚C –27˚C. Silkworms were fed an artificial diet, Silkmate 2S, containing antibiotics purchased from Ehime-Sanshu Co., Ltd. Fifth-instar larvae were used for the infection experiments.

### *In vivo* melanization assay

An *in vivo* melanization assay was performed as previously described [22], with slight modifications. The silkworm injection experiments were performed as previously described [29]. Fifth instar silkworm larvae were fed an artificial diet (1.5 g Silkmate 2S; Ehime-Sanshu Co., Ltd) overnight. A 50-µL suspension of bacterial samples was injected into the silkworm hemolymph with a 1-ml tuberculin syringe (Terumo Medical Corporation, Tokyo, Japan). The silkworms were maintained at 37˚C for 3 h. Hemolymph was collected from the larvae through a cut on the first proleg as described previously [30]. The silkworm hemolymph (50 µL) was mixed with 50 µL of physiologic saline solution (0.9% NaCl: PSS). Absorbance at 490 nm was measured using a microplate reader (iMark™ microplate reader; Bio-Rad Laboratories Inc., Hercules, CA, USA). Each experiment was performed at least twice to check reproducibility.

### Protease treatment

Autoclaved *S. aureus* cells [AC] were diluted with phosphate-buffered saline (PBS) to an absorbance at 600 nm ($A_{600}$) = 3 in 1 mL, and 50 µL protease K (0.75 AU/ml) was added. After incubation for 1 h at 50˚C, the samples were centrifuged at 15,000 rpm for 10 min at room temperature. The precipitate was suspended in PSS (1 mL), and the remaining enzymes were inactivated by incubation at 80˚C for 30 min. The samples were centrifuged at 15,000 rpm for 10 min at room temperature and the precipitate was diluted with PSS to $A_{600}$ = 1 to obtain the precipitate sample.

### Statistical analysis

Statistical differences between groups were analyzed using the Student's *t*-test, the Tukey's test or the Tukey-Kramer test. The Student's *t*-test was used to assess whether the two groups were statistically significantly different. The Tukey's test and the Tukey-Kramer test were used to assess whether the multiple groups were statistically significantly different. Each experiment was performed at least twice. A *P* value of less than 0.05 was considered statistically significant.

## Results

### Comparison of melanization-inducing activity between heat-killed *C. acnes* and *S. aureus*

The injection of heat-killed *C. acnes* cells obtained by autoclaving causes silkworm hemolymph melanization [22]. We examined whether heat-killed *C. acnes* and *S. aureus* cells differed in their ability to induce silkworm hemolymph melanization. The experimental scheme for silkworm hemolymph melanization is shown in Fig 1. Melanization was induced by injection of autoclave-treated *C. acnes*, but not by that of autoclave-treated *S. aureus* (Fig 2A and 2B). The result suggests that the silkworm hemolymph melanization activity induced by heat-killed *S. aureus* is lower than that by heat-killed *C. acnes*.

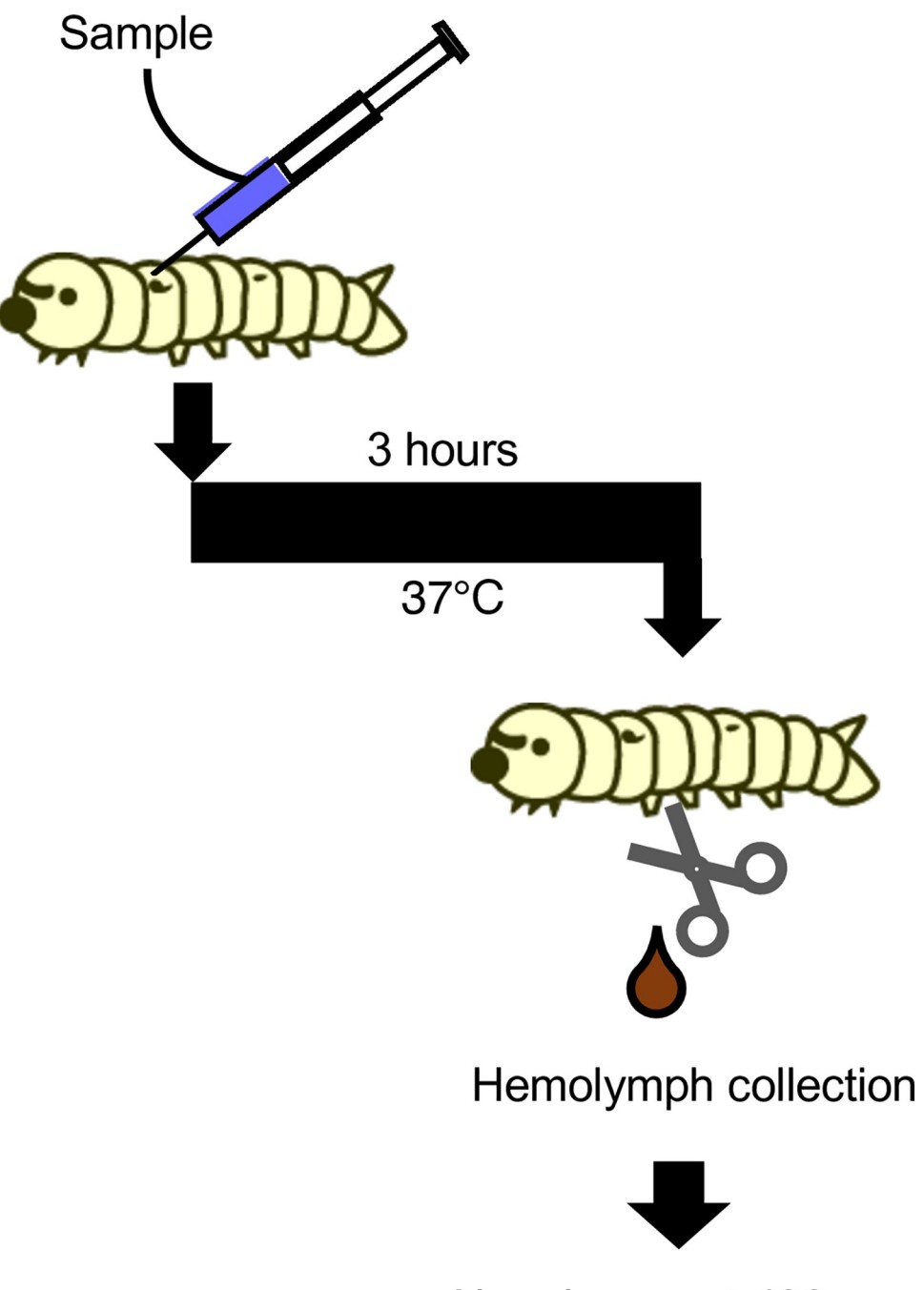

**Fig 1. Evaluation system for inducing silkworm hemolymph melanization using bacterial samples.** Illustration of an experimental method to determine silkworm hemolymph melanization. Sample solution was injected into the silkworm hemolymph. The silkworms were maintained at 37˚C for 3 h. The hemolymph was collected from the larvae through a cut on the first proleg. Absorbance was measured at 490 nm.

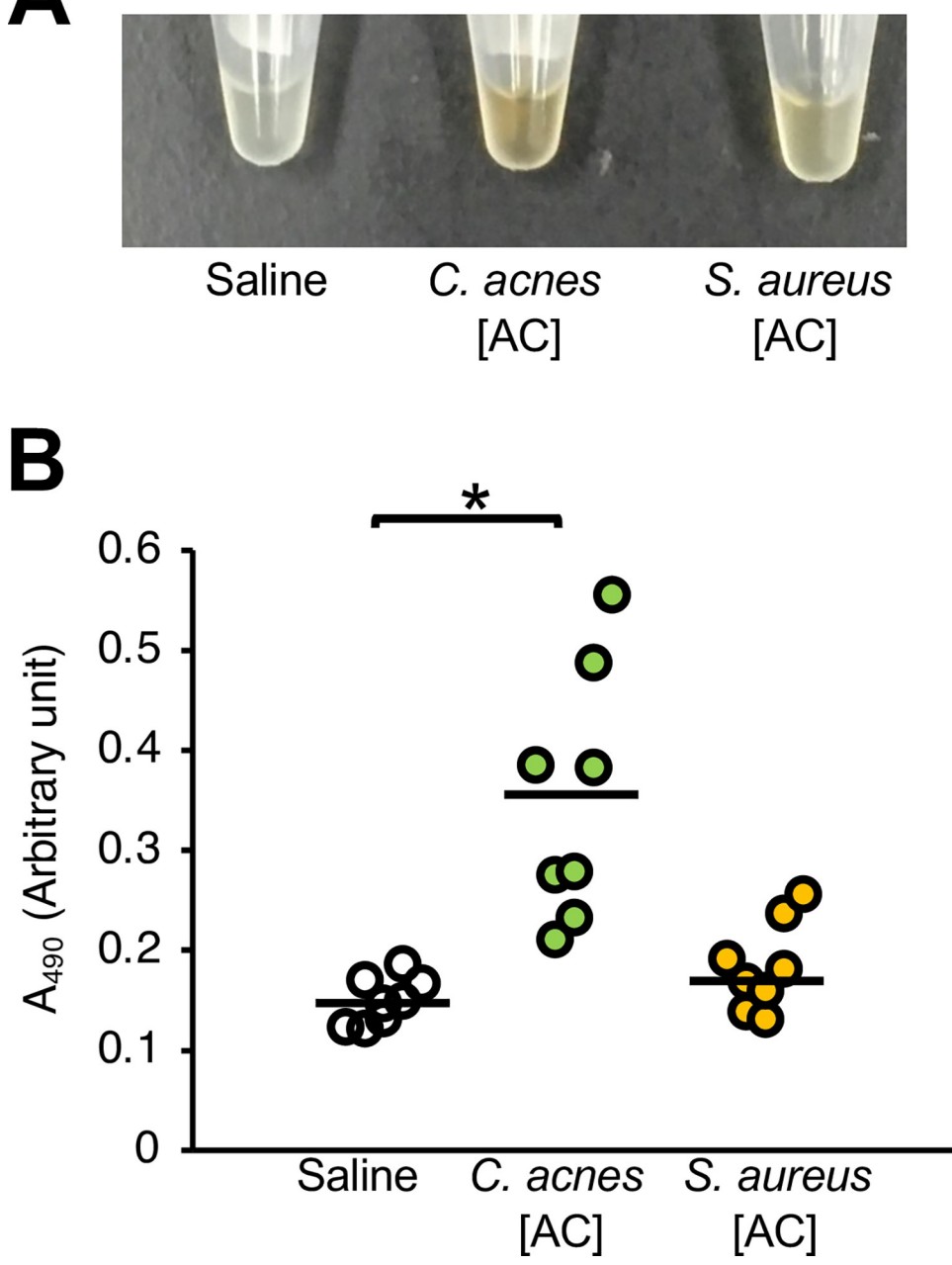

**Fig 2. Comparison of silkworm hemolymph melanization induced by heat-killed *C. acnes* and *S. aureus*. (A, B)** Sample solution was injected to silkworms and hemolymph was collected at 3 hours after injection. Samples were saline (Saline), heat-killed *C. acnes* cell suspension (*C. acnes*; 1 x 10$^8$ cells/larva), or heat-killed *S. aureus* cell suspension (*S. aureus*; 1 x 10$^8$ cells/larva). (**A**) Photograph. (**B**) Absorbance at 490 nm (A$_{490}$). n = 8/group. Statistically significant differences between groups were evaluated using the Tukey's test. *$P < 0.05$.

### Increased melanization-inducing activity of *S. aureus* after treatment with *C. acnes* culture supernatant

Silkworm hemolymph melanization was increased by injecting silkworms with *S. aureus* [AC] treated with *C. acnes* CS (Fig 3). We confirmed that *C. acnes* CS alone did not induce silkworm

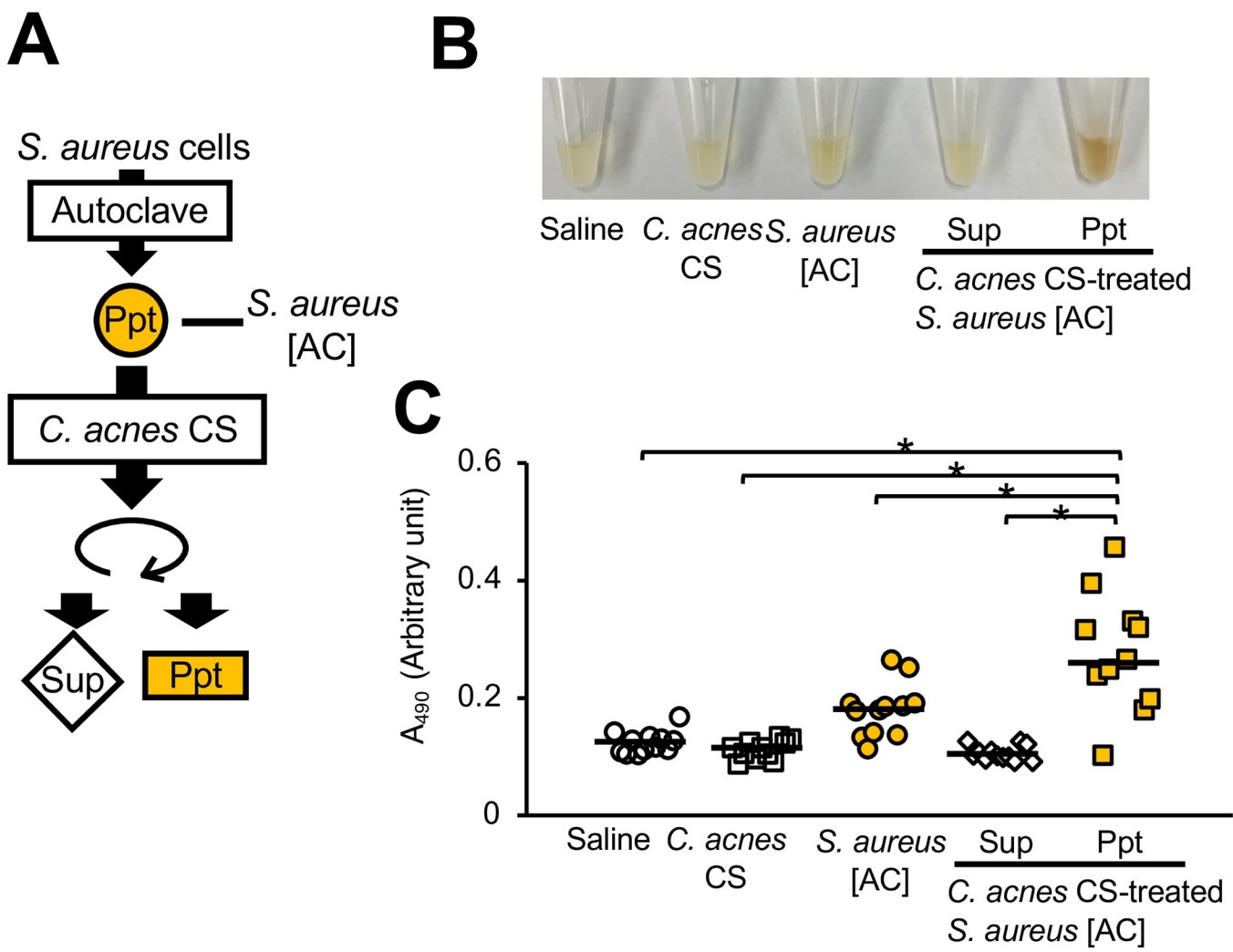

**Fig 3. Silkworm hemolymph melanization was induced by injecting heat-killed *S. aureus* treated with *C. acnes* culture supernatant.** (**A**) Preparation of *C. acnes* culture supernatant (CS)-treated *S. aureus* [AC]. The *S. aureus* [AC] fraction was treated with *C. acnes* CS at 37°C for 24 h. (**B**, **C**) Sample solution was injected to silkworms and hemolymph was collected at 3 hours after injection. Samples were saline (Saline), *C. acnes* CS, *S. aureus* [AC], *C. acnes* CS-treated *S. aureus* [AC] supernatant (Sup), or precipitate (Ppt). (**B**) Photograph. (**C**) Absorbance at 490 nm ($A_{490}$). n = 11-12/group. Statistically significant differences between groups were evaluated using the Tukey-Kramer test. *$P < 0.05$.

hemolymph melanization (Fig 3). These results suggest that the substances produced by *C. acnes* induced the silkworm hemolymph melanization activity of *S. aureus*.

## Silkworm hemolymph melanization by acid-treated *S. aureus* cells

*C. acnes* produces short-chain fatty acids such as propionic acid [13], which affect the pH in the environment [3, 31]. The pH of the *C. acnes* CS used in this study was lower than that of the preculture medium (Table 1). Therefore, we tested whether silkworm hemolymph melanization by *S. aureus* [AC] was induced by pretreatment with propionic acid. Propionic acid-treated *S. aureus* [AC] induced silkworm hemolymph melanization (Fig 4). Next, we examined whether acids other than propionic acid increased the silkworm hemolymph melanization activity of *S. aureus* [AC]. Compared with untreated *S. aureus* [AC], *S. aureus* [AC] treated with acetic acid, lactic acid, or hydrochloric acid also exhibited higher silkworm hemolymph

**Table 1. Decrease in the pH of medium by the addition of *C. acnes* culture.**

|  | pH | |
|---|---|---|
|  | **Before** | **After*** |
| *C. acnes* | 6.2 | 4.5 |

* *C. acnes* (8 x 10⁹ cells) was added to 10 mL of tryptic soy broth + 2% glucose medium and incubated at 37˚C for 6 days under anaerobic conditions.

melanization activity (Fig 5). On the other hand, administration of these short-chain fatty acid solutions did not affect melanization of the silkworm hemolymph (S1 Fig in S1 File). Furthermore, injections of *S. aureus* [AC] treated with low-pH solutions adjusted by hydrochloric acid at 37˚C for 24 h also enhanced silkworm hemolymph melanization in a pH-dependent manner, with more melanization occurring at a lower pH (Fig 6). These results suggest that acid treatment induces silkworm hemolymph melanization activity by *S. aureus*.

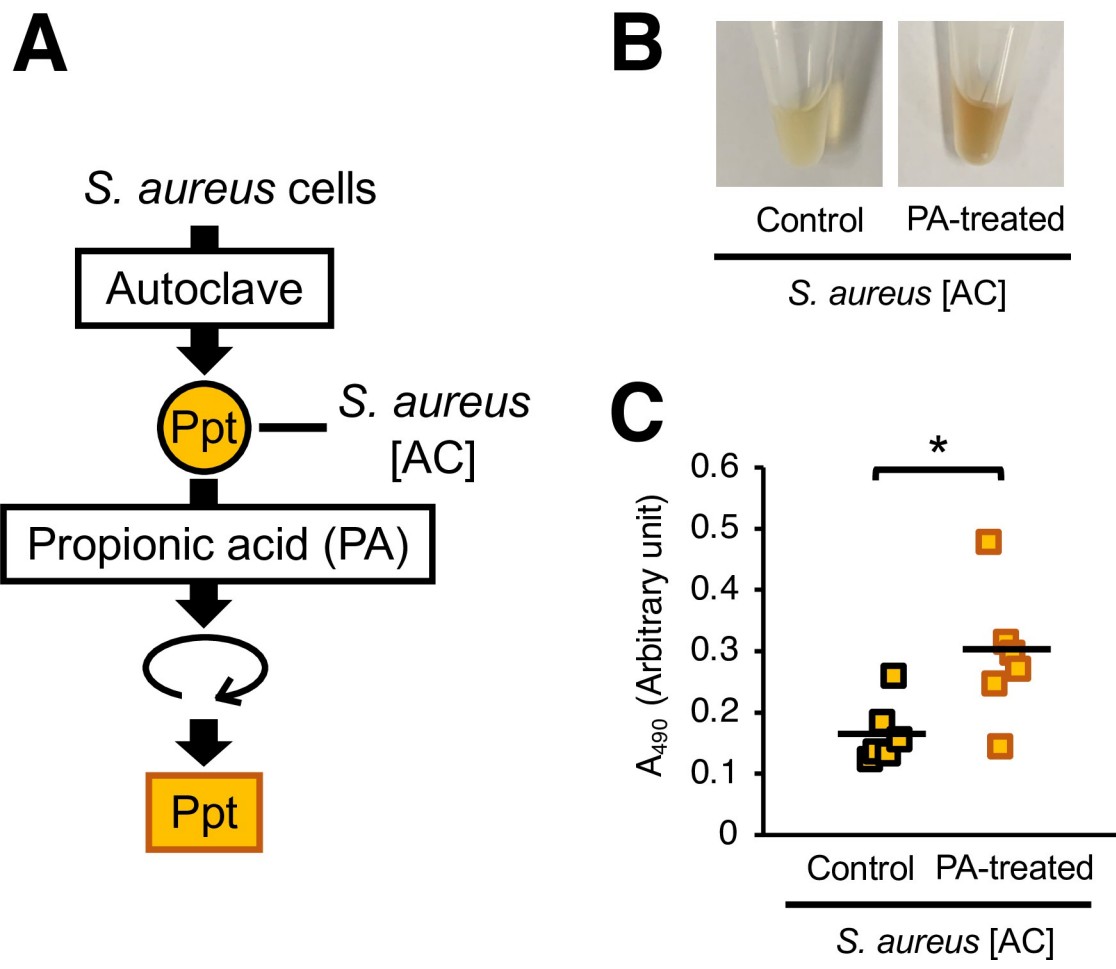

**Fig 4. Silkworm hemolymph melanization induced by the injection of heat-killed *S. aureus* treated with propionic acid. (A)** Preparation of propionic acid (PA)-treated *S. aureus* [AC]. The *S. aureus* [AC] fraction was treated with propionic acid (1%: 130 mM) at 37˚C for 24 h. (**B, C**) Sample solution was injected to silkworms and hemolymph was collected at 3 hours after injection. Samples were *S. aureus* [AC] or propionic acid (130 mM)-treated *S. aureus* [AC] precipitates (Ppt). (**B**) Photograph. (**C**) Absorbance at 490 nm ($A_{490}$). n = 6/group. Statistically significant differences between groups were evaluated using the Tukey-Kramer test. *$P < 0.05$.

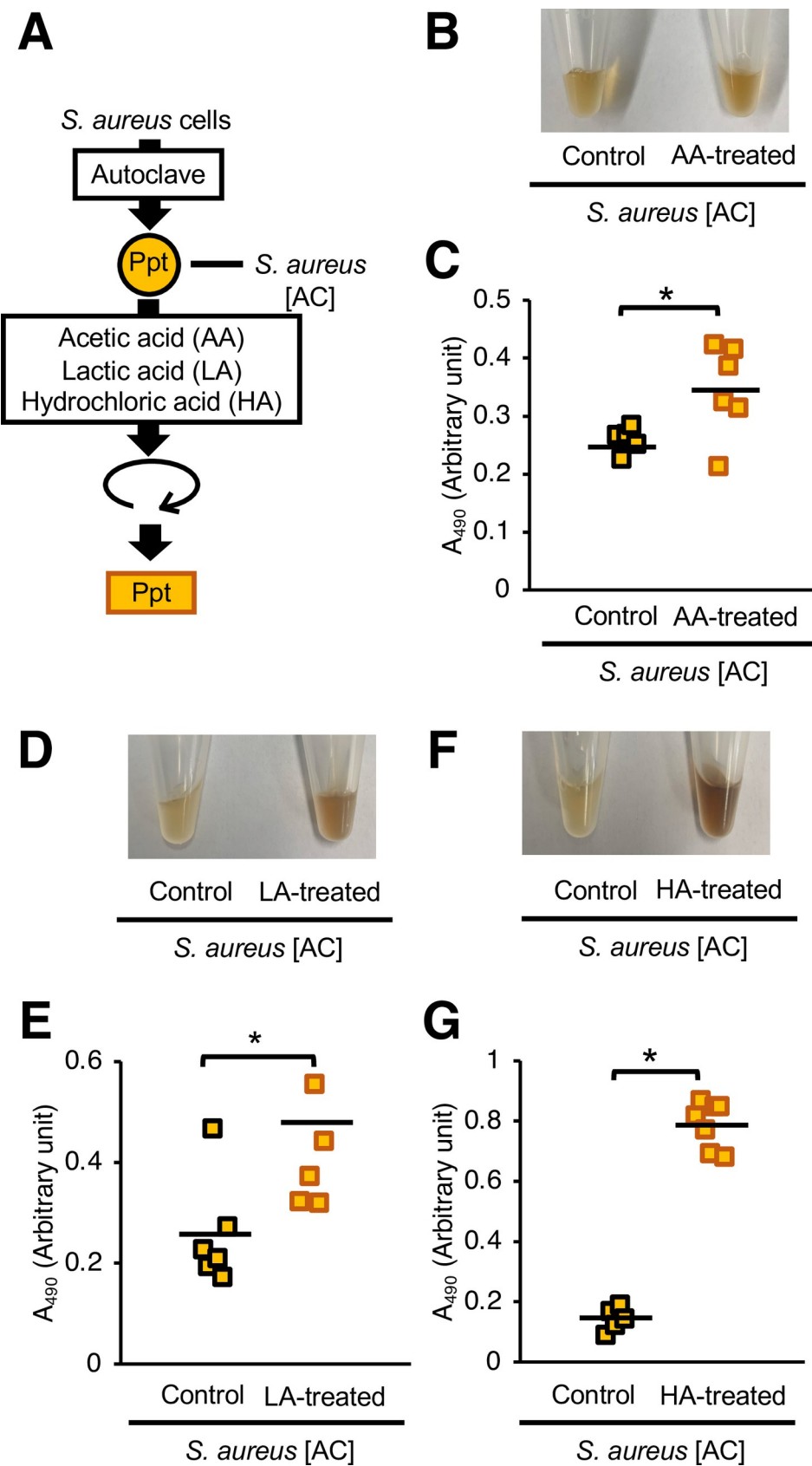

**Fig 5. Effects of acid treatment of heat-killed *S. aureus* on silkworm hemolymph melanization.** (**A**) Preparation of the acid-treated *S. aureus* [AC]. The *S. aureus* [AC] fraction was treated with acetic acid (AA) (130 mM), lactic acid (LA) (130 mM), or hydrochloric acid (HA) (130 mM) at 37°C for 24 h. (**B**, **C**) Sample solution was injected to silkworms and hemolymph was collected at 3 hours after injection. Samples were *S. aureus* [AC], AA, LA or HA-treated *S. aureus* [AC] precipitates (Ppt). (**B**) Photograph. (**C**) Absorbance at 490 nm (A$_{490}$). n = 5-7/group. Statistically significant differences between groups were evaluated using the Student's *t*-test. *$P < 0.05$.

## Effect of protease treatment on the induction of silkworm hemolymph melanization by *S. aureus*

*S. aureus* lipoproteins bind to mammalian TLR2 and activate innate immunity [6, 7]. Therefore, we hypothesized that a protein in *S. aureus* [AC] was responsible for this activity. Melanization of the silkworm hemolymph by *S. aureus* [AC] treated with low pH was decreased by pretreating the *S. aureus* [AC] with a protease (Fig 7). On the other hand, administration of

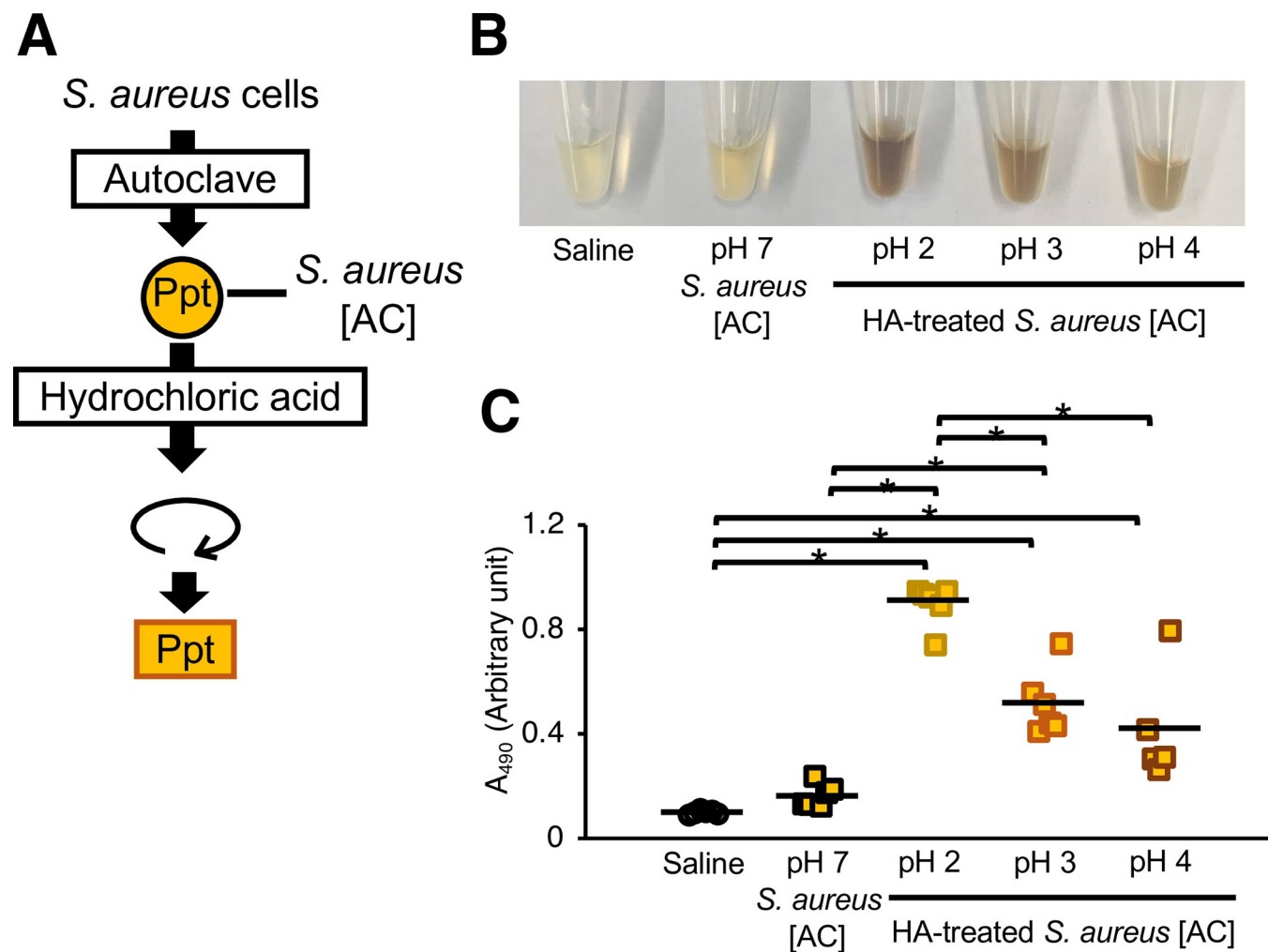

**Fig 6. Effects of low-pH treatments of heat-killed *S. aureus* on silkworm hemolymph melanization.** (**A**) Preparation of the acid-treated *S. aureus* [AC]. The *S. aureus* [AC] fraction was treated with saline or hydrochloric acid solution adjusted to pH 2–7 at 37°C for 24 h. (**B**, **C**) Sample solution was injected to silkworms and hemolymph was collected at 3 hours after injection. Samples were *S. aureus* [AC] or low pH-treated *S. aureus* [AC] precipitate (Ppt). (**B**) Photograph. (**C**) Absorbance at 490 nm (A$_{490}$). n = 5-6/group. Statistically significant differences between groups were evaluated using the Tukey-Kramer test. *$P < 0.05$.

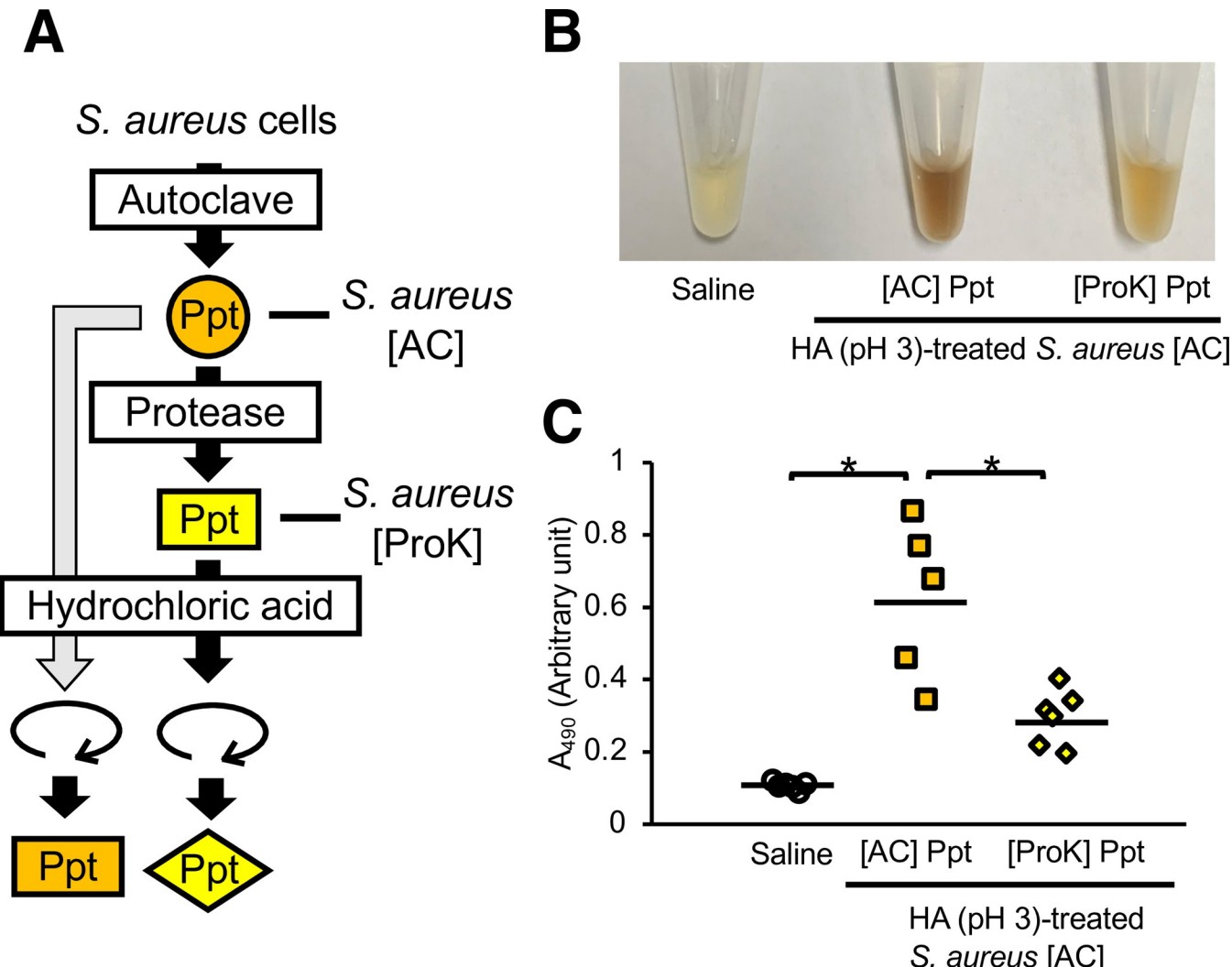

**Fig 7. Effect of protease treatment of heat-killed *S. aureus* on silkworm hemolymph melanization.** (**A**) Preparation of the acid-treated *S. aureus* [AC] with protease treatment. *S. aureus* [AC] was treated with protease K. Protease K-treated *S. aureus* [AC] was further treated with hydrochloric acid solution adjusted to pH 3 at 37°C for 7 days. (**B, C**) Sample solution was injected to silkworms and hemolymph was collected at 3 hours after injection. Samples were saline (Saline), acid-treated *S. aureus* [AC] precipitate ([AC] Ppt), or acid-treated protease K-treated *S. aureus* [AC] precipitate ([ProK] Ppt). (**B**) Photograph. (**C**) Absorbance at 490 nm ($A_{490}$). n = 5-6/group. Statistically significant differences between groups were evaluated using the Tukey-Kramer test. *$P < 0.05$.

protease solution did not affect melanization of the silkworm hemolymph (S1 Fig in S1 File). These results suggest that *S. aureus* proteins are involved in the induction of silkworm hemolymph melanization by acid-treated *S. aureus*.

## Discussion

Acid treatment of *S. aureus* stimulates innate immune activity in silkworms, and the innate immune activating substances of *S. aureus* may be proteins. Our findings suggest that acid-treated *S. aureus* induces excess host immunity, which may cause immune-active diseases, such as folliculitis.

*S. aureus* and *C. acnes* are gram-positive bacteria present in the human skin that cause inflammatory diseases such as folliculitis [2]. We hypothesized that an interaction between *S. aureus* and *C. acnes* within the hair follicle may influence the induction of innate immune

responses by *S. aureus*. In the present study, we found that acidic substances released by *C. acnes* affected the immune-inducing activity of *S. aureus*. Further studies should be conducted in mammalian models to confirm and develop these findings.

On the human skin, various bacteria, including *C. acnes* and *S. aureus*, produce short-chain fatty acids such as lactic acid [32]. Moreover, *S. aureus* and *C. acnes* produce lipases that produce free fatty acids from sebum [13], which can decrease the pH of the skin. On the other hand, bacteria such as *Delftia acidovorans*, a gram-negative bacterium on human skin, produce ammonia that increases the pH [33]. Therefore, the balance of the skin microbiome may affect changes in skin pH. Weak acidity caused by alterations in the skin microbiome may induce an innate immune response by *S. aureus*. Proteins of *S. aureus* may be involved in the immune activity induced by acid-treated *S. aureus* in the silkworm evaluation system. The conformational changes of proteins responsible for inducing innate immunity by acid treatment, however, remain unknown. We assumed that *S. aureus* lipoproteins treated with acids lead to conformational changes, and that acid-treated lipoproteins are easily recognized by Toll receptors, which are involved in innate immunity in silkworms. Preventing the acid-enhanced induction of innate immune activation by *S. aureus* may contribute to inhibiting the onset of inflammatory skin diseases such as folliculitis.

As a limitation of this study, the effects of acid-treated *S. aureus* on inflammation in individual mammals were not evaluated. It is also unclear whether the effects of an acidic pH on the skin microbiome are related to immune induction. Furthermore, the responsible proteins of *S. aureus* have not yet been identified. The effects of acid treatment and the identification of the responsible proteins are important issues for future research.

## Conclusion

Acid-treated *S. aureus* induces innate immunity in silkworms, and *S. aureus* proteins may be the responsible factors.

## Supporting information

**S1 File.**
(DOCX)

**S1 Dataset. Datasets included in this study.**
(XLSX)

## Acknowledgments

We thank Yu Sugiyama, Sachi Koganesawa, and Hiromi Kanai (Meiji Pharmaceutical University) for technical assistance rearing the silkworms. We also thank the editors of SciTechEdit International LLC (Highlands Ranch, CO, USA) for providing editorial support during the production of this manuscript.

## Author Contributions

**Conceptualization:** Yasuhiko Matsumoto.

**Data curation:** Yasuhiko Matsumoto, Eri Sato.

**Formal analysis:** Yasuhiko Matsumoto, Eri Sato.

**Funding acquisition:** Yasuhiko Matsumoto.

**Investigation:** Yasuhiko Matsumoto, Eri Sato.

**Methodology:** Yasuhiko Matsumoto, Eri Sato.

**Project administration:** Yasuhiko Matsumoto, Takashi Sugita.

**Supervision:** Yasuhiko Matsumoto.

**Visualization:** Yasuhiko Matsumoto, Eri Sato.

**Writing – original draft:** Yasuhiko Matsumoto.

**Writing – review & editing:** Yasuhiko Matsumoto, Takashi Sugita.

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
