## [Decision Letter · Decision Letter 0]

24 Mar 2024

PONE-D-24-03321Acid-treated Staphylococcus aureus induces acute silkworm hemolymph melanizationPLOS ONE

Dear Dr. Matsumoto,

Thank you for submitting your manuscript to PLOS ONE. After careful consideration, we feel that it has merit but does not fully meet PLOS ONE’s publication criteria as it currently stands. Therefore, we invite you to submit a revised version of the manuscript that addresses the points raised during the review process.

We look forward to receiving your revised manuscript.

Kind regards,

Jian Xu, Ph.D.

Academic Editor

PLOS ONE

Journal Requirements:

**Additional Editor Comments:**

1. Fig. 1: Absorbance (at) 490 nm; ℃ is not in correct font style.

2. Fig. 2: legend for 2B is incorrect.

3. Figs.2-7: Decrease the size of symbols a bit used for data point.

4. Method section must be carefully checked to ensure the reproducibility. 

4. Please explain the methods for statistical analyses and reasons

5. Please discuss more on the mechanism based on the findings of this study. 

6. The limitations of this study should also be clearly indicated in an additional section. 

Reviewers' comments:

Reviewer's Responses to Questions

**Comments to the Author**

1. Is the manuscript technically sound, and do the data support the conclusions?

Reviewer #1: Yes

2. Has the statistical analysis been performed appropriately and rigorously? 

Reviewer #1: Yes

3. Have the authors made all data underlying the findings in their manuscript fully available?

Reviewer #1: Yes

4. Is the manuscript presented in an intelligible fashion and written in standard English?

Reviewer #1: Yes

5. Review Comments to the Author

Reviewer #1: Silkworms are a suitable model for evaluating the infection and pathogenic mechanism of pathogenic microorganisms. In the manuscript, Matsumoto et al. investigated the hemolymph melanization induced by Staphylococcus aureus in the silkworm model under the participation of Cutibacterium acnes. S. aureus and C. acnes are Gram-positive bacteria present on human skin that can cause inflammatory skin diseases. It is valuable to evaluate the effect of C. acnes on host immunity induced by S. aureus. They found that acidic substances released by C. acnes affect the hemolymph melanization reaction induced by S. aureus. The silkworm hemolymph melanization activity of S. aureus treated with hydrochloric acid was inhibited by protease treatment of S. aureus. I think these results are a beneficial addition to understanding the impact of microbial interactions on the human health of the skin surface. However, there are still some writing logic disorder and incomplete descriptions in this manuscript, some data lack persuasiveness, and the author needs to spend some time carefully revising it.

1.In Figures 4 and 5, the authors should investigate the levels of hemolymph melanization caused by PA, LA, AA, HA, and proteases under the same conditions.

2.The research results did not show or imply an effect of S. aureus on C. acnes, and the author did not indicate changes in C. acnes during S. aureus-induced folliculitis in the literature. Therefore, further experimental evidence is clearly needed to support the statement that "Preventing the acid-enhanced induction of innate immune activation by S. aureus may contribute to inhibiting the onset of inflammatory skin diseases such as folliculitis."

3.The introduction is not smoothly written. The third paragraph (lines 60-67) should come after the first paragraph (lines 38-46).

4.In the Discussion section, there should be further strengthened on how this study deepens our understanding of inflammatory diseases caused by S. aureus. For instance, there exists a vast microbial community on the human skin surface, with S. aureus being an opportunistic pathogen that interacts with other microbes besides C. acnes.

5.The statement "Preventing the acid-enhanced induction of innate immune activation by S. aureus may contribute to inhibiting the onset of inflammatory skin diseases such as folliculitis" in the conclusion should be placed in the discussion."

6.The authors need to provide the silkworm strains used for this study in the Materials and Methods section.

7.Spelling issue. For example, the symbol "℃" in the caption of Figure 1.

8.Italics. Line 118, The "P" should be italicized.

6. PLOS authors have the option to publish the peer review history of their article (what does this mean?). If published, this will include your full peer review and any attached files.

Reviewer #1: No

---

## [Author Response · Author response to Decision Letter 0]

27 Mar 2024

Additional Editor Comments:

1. Fig. 1: Absorbance (at) 490 nm; ℃ is not in correct font style.

According to the editor’s comment, these font styles were revised (Fig. 1). 

2. Fig. 2: legend for 2B is incorrect.

According to the editor’s comment, the Figure legend of Fig. 2 was revised (Page 8, lines 146-147). 

3. Figs.2-7: Decrease the size of symbols a bit used for data point.

According to the editor’s comment, the size of symbols in Fig. 2-7 was decreased.

4. Method section must be carefully checked to ensure the reproducibility. 

According to the editor’s comment, we added the sentence in the Method section of the revised manuscript (Page 6, line 106).

[Page 6, line 106]

Each experiment was performed at least twice to check reproducibility.

5. Please explain the methods for statistical analyses and reasons

According to the editor’s comment, we described the reasons to use these statistical analyses in the Method section of the revised manuscript (Page 6, lines 119-121).

[Page 6, lines 119-121]

The Student's t-test was used to assess whether the two groups were statistically significantly different. The Tukey’s test and the Tukey-Kramer test were used to assess whether the multiple groups were statistically significantly different.

6. Please discuss more on the mechanism based on the findings of this study. 

Following the editor’s comment, we added the sentences in the Discussion section of the revised manuscript (Page 12, lines 245-250).

[Page 12, lines 245-250]

Proteins of S. aureus may be involved in the immune activity induced by acid-treated S. aureus in the silkworm evaluation system. The conformational changes of proteins responsible for inducing innate immunity by acid treatment, however, remain unknown. We assumed that S. aureus lipoproteins treated with acids lead to conformational changes, and that acid-treated lipoproteins are easily recognized by Toll receptors, which are involved in innate immunity in silkworms. Identification of the responsible proteins and the effects of acid treatment are important issues for future research.

7. The limitations of this study should also be clearly indicated in an additional section.

According to the editor’s comment, we described the limitations in the new paragraph in the Discussion section of the revised manuscript (Page 12, lines 252-256).

[Page 12, lines 251-255]

As a limitation of this study, the effects of acid-treated S. aureus on inflammation in individual mammals were not evaluated. It is also unclear whether the effects of an acidic pH on the skin microbiome are related to immune induction. Furthermore, the responsible proteins of S. aureus have not yet been identified. The effects of acid treatment and the identification of the responsible proteins are important issues for future research.

Reviewer #1: Silkworms are a suitable model for evaluating the infection and pathogenic mechanism of pathogenic microorganisms. In the manuscript, Matsumoto et al. investigated the hemolymph melanization induced by Staphylococcus aureus in the silkworm model under the participation of Cutibacterium acnes. S. aureus and C. acnes are Gram-positive bacteria present on human skin that can cause inflammatory skin diseases. It is valuable to evaluate the effect of C. acnes on host immunity induced by S. aureus. They found that acidic substances released by C. acnes affect the hemolymph melanization reaction induced by S. aureus. The silkworm hemolymph melanization activity of S. aureus treated with hydrochloric acid was inhibited by protease treatment of S. aureus. I think these results are a beneficial addition to understanding the impact of microbial interactions on the human health of the skin surface. However, there are still some writing logic disorder and incomplete descriptions in this manuscript, some data lack persuasiveness, and the author needs to spend some time carefully revising it.

1.In Figures 4 and 5, the authors should investigate the levels of hemolymph melanization caused by PA, LA, AA, HA, and proteases under the same conditions.

According to the reviewer’s comment, we performed the new experiments (Supplementary Fig. S1). The administration of PA, LA, AA, HA, and proteases did not affect silkworm hemolymph melinization. The Supplementary Fig. 1 was presented in the S1 File. The results were described in the Results section of the revised manuscript (Page 9, lines 174-176, page 11, lines 214-215). 

[Page 9, lines 174-176]

On the other hand, administration of these short-chain fatty acid solutions did not affect melanization of the silkworm hemolymph (S1 Fig in S1 File).

[Page 11, lines 214-215]

On the other hand, administration of protease solution did not affect melanization of the silkworm hemolymph (S1 Fig in S1 File).

2.The research results did not show or imply an effect of S. aureus on C. acnes, and the author did not indicate changes in C. acnes during S. aureus-induced folliculitis in the literature. Therefore, further experimental evidence is clearly needed to support the statement that "Preventing the acid-enhanced induction of innate immune activation by S. aureus may contribute to inhibiting the onset of inflammatory skin diseases such as folliculitis."

Following the reviewer’s comment, we added the limitations in this study in the revised manuscript (Page 12, lines 252-256). 

[Page 12, lines 252-256]

As a limitation of this study, the effects of acid-treated S. aureus on inflammation in individual mammals were not evaluated. It is also unclear whether the effects of an acidic pH on the skin microbiome are related to immune induction. Furthermore, the responsible proteins of S. aureus have not yet been identified. The effects of acid treatment and the identification of the responsible proteins are important issues for future research.

3.The introduction is not smoothly written. The third paragraph (lines 60-67) should come after the first paragraph (lines 38-46).

According to the reviewer’s comment, we have changed the paragraph in the revised manuscript (Page 3, lines 47-54). 

4.In the Discussion section, there should be further strengthened on how this study deepens our understanding of inflammatory diseases caused by S. aureus. For instance, there exists a vast microbial community on the human skin surface, with S. aureus being an opportunistic pathogen that interacts with other microbes besides C. acnes.

According to the reviewer’s comment, we changed the sentences in the Discussion section of the revised manuscript (Page 12, lines 239-245). The effects of Delftia acidovorans was described. 

[Page 12, lines 239-245]

On the human skin, various bacteria, including C. acnes and S. aureus, produce short-chain fatty acids such as lactic acid [32]. Moreover, S. aureus and C. acnes produce lipases that produce free fatty acids from sebum [13], which can decrease the pH of the skin. On the other hand, bacteria such as Delftia acidovorans, a gram-negative bacterium on human skin, produce ammonia that increases the pH [33]. Therefore, the balance of the skin microbiome may affect changes in skin pH. Weak acidity caused by alterations in the skin microbiome may induce an innate immune response by S. aureus.

5.The statement "Preventing the acid-enhanced induction of innate immune activation by S. aureus may contribute to inhibiting the onset of inflammatory skin diseases such as folliculitis" in the conclusion should be placed in the discussion."

Following the reviewer’s comment, we moved the sentence in the Discussion section of the revised manuscript (Page 12, lines 250-251). 

6.The authors need to provide the silkworm strains used for this study in the Materials and Methods section.

Following the reviewer’s comment, we added the information of the silkworm strain in the Method section in the revised manuscript (Page 5, lines 91-92).

[Page 5, lines 91-92]

Silkworm eggs (Hu Yo × Tukuba Ne) were purchased from Ehime-Sanshu Co. Ltd. (Ehime, Japan), disinfected, and hatched at 25℃ –27℃.

7.Spelling issue. For example, the symbol "℃" in the caption of Figure 1.

According to the reviewer’s comment, the symbol in the Figure legend of Figure 1 were revised (Page 7, line 138). Moreover, the symbol "℃" in the Figure legends of Figure 3-7 also revised (Page 8, line 160, page 9, line 185, page 10, line 195, line, 203, and page 11, line 223). 

8.Italics. Line 118, The "P" should be italicized.

According to the reviewer’s comment, the style of “P” in the Method section were changed to italic (Page 6, line 122).

---

## [Decision Letter · Decision Letter 1]

8 Apr 2024

PONE-D-24-03321R1Acid-treated Staphylococcus aureus induces acute silkworm hemolymph melanizationPLOS ONE

Dear Dr. Matsumoto,

Thank you for submitting your manuscript to PLOS ONE. After careful consideration, we feel that it has merit but does not fully meet PLOS ONE’s publication criteria as it currently stands. Therefore, we invite you to submit a revised version of the manuscript that addresses the points raised during the review process.

 Please submit your revised manuscript by May 23 2024 11:59PM. If you will need more time than this to complete your revisions, please reply to this message or contact the journal office at plosone@plos.org. Please include the following items when submitting your revised manuscript:A rebuttal letter that responds to each point raised by the academic editor and reviewer(s). You should upload this letter as a separate file labeled 'Response to Reviewers'.A marked-up copy of your manuscript that highlights changes made to the original version. You should upload this as a separate file labeled 'Revised Manuscript with Track Changes'.An unmarked version of your revised paper without tracked changes. You should upload this as a separate file labeled 'Manuscript'.If applicable, we recommend that you deposit your laboratory protocols in protocols.io to enhance the reproducibility of your results. Protocols.io assigns your protocol its own identifier (DOI) so that it can be cited independently in the future. For instructions see: https://journals.plos.org/plosone/s/submission-guidelines#loc-laboratory-protocols. Additionally, PLOS ONE offers an option for publishing peer-reviewed Lab Protocol articles, which describe protocols hosted on protocols.io. Read more information on sharing protocols at https://plos.org/protocols?utm_medium=editorial-email&utm_source=authorletters&utm_campaign=protocols.

We look forward to receiving your revised manuscript.

Kind regards,

Jian Xu, Ph.D.

Academic Editor

PLOS ONE

Journal Requirements:

**Additional Editor Comments:**

The reviewer and editor feel that your modifications are satisfactory and they pointed out only one minor change of your manuscript (Line 47). Please revise and make a final version of your manuscript to make sure all parts are updated.

Reviewers' comments:

Reviewer's Responses to Questions

**Comments to the Author**

1. If the authors have adequately addressed your comments raised in a previous round of review and you feel that this manuscript is now acceptable for publication, you may indicate that here to bypass the “Comments to the Author” section, enter your conflict of interest statement in the “Confidential to Editor” section, and submit your "Accept" recommendation.

Reviewer #1: (No Response)

2. Is the manuscript technically sound, and do the data support the conclusions?

Reviewer #1: (No Response)

3. Has the statistical analysis been performed appropriately and rigorously? 

Reviewer #1: (No Response)

4. Have the authors made all data underlying the findings in their manuscript fully available?

Reviewer #1: (No Response)

5. Is the manuscript presented in an intelligible fashion and written in standard English?

Reviewer #1: (No Response)

6. Review Comments to the Author

Reviewer #1: Abbreviations should indicate their full name when first appearing in the main text. Line 47, C. Acnes.

7. PLOS authors have the option to publish the peer review history of their article (what does this mean?). If published, this will include your full peer review and any attached files.

Reviewer #1: No

---

## [Author Response · Author response to Decision Letter 1]

8 Apr 2024

Journal Requirements:

According to the editor’s comment, we checked again the reference list. The reference list was not changed.

Additional Editor Comments:

The reviewer and editor feel that your modifications are satisfactory and they pointed out only one minor change of your manuscript (Line 47). Please revise and make a final version of your manuscript to make sure all parts are updated.

According to the editor’s comment, we changed the words of the revised manuscript (Line 47).

[Line 47]

Cutibacterium acnes, a gram-positive bacterium on the human skin,

Reviewer Comments:

Reviewer #1: Abbreviations should indicate their full name when first appearing in the main text. Line 47, C. acnes.

According to the editor’s comment, we changed the words of the revised manuscript (Line 47).

[Line 47]

Cutibacterium acnes, a gram-positive bacterium on the human skin,

---

## [Editor Report · Decision Letter 2]

10 Apr 2024

Acid-treated Staphylococcus aureus induces acute silkworm hemolymph melanization

PONE-D-24-03321R2

Dear Dr. Matsumoto,

We’re pleased to inform you that your manuscript has been judged scientifically suitable for publication and will be formally accepted for publication once it meets all outstanding technical requirements.

Kind regards,

Jian Xu, Ph.D.

Academic Editor

PLOS ONE
---

## [Editor Report · Acceptance letter]

17 May 2024

PONE-D-24-03321R2 

PLOS ONE

Dear Dr. Matsumoto, 

I'm pleased to inform you that your manuscript has been deemed suitable for publication in PLOS ONE. Congratulations! Your manuscript is now being handed over to our production team.

Kind regards, 

on behalf of

Dr. Jian Xu 

Academic Editor

PLOS ONE